# OpenReview forum: "AgentFold: Long-Horizon Web Agents with Proactive Context Folding"
_ICLR.cc/2026/Conference — ICLR 2026 Poster_

### Official Review · Reviewer_mj6d · 2025-10-25

**Soundness:** 3
**Presentation:** 2
**Contribution:** 3
**Rating:** 4
**Confidence:** 4

**Summary:**

This paper addresses the important long-history handling issue among agents, by proposing AgentFold, which consolidates the latest interactions into fine- and coarse-grained memory. Experiments across different benchmarks demonstrate its empirical advantages among open-source LMs.

**Strengths:**

**1. Importance of Problem and Novel Approach.**
> This paper tackles an important and prevailing issue among agents – their inability to process long contexts as tasks grow more complex. This paper proposes a new angle to actively manage, instead of passively log, past interaction history.

**2. Substantial Experimental Improvements.**
> Results in Table 1 show the empirical effectiveness of the proposed AgentFold method, among open-source models. Further analysis of the number of turns provides additional insights.

**Weaknesses:**

**1. Lack of Clarity in Method Description.**
> While Section 3 describes the two condensation methods at a high level, it is still unclear what counts as a “fine-grained” summary (in granular condensation) and what counts as a “coarse-grained” summary (dense condensation). It would be helpful if Figure 2 (or an additional figure) could show a few concrete examples of both.

**2. Lack of Hyper-Parameter Choices.**
> Several values could be critical to the empirical performance of the proposed method, but are not reported, justified, or hard to find. These values include: (i) number of steps to trigger consolidation, (ii) number of fine- and coarse-grained memory entries to maintain, (iii) number of steps of the “latest interaction”. Further, if these values need to be adjusted when applying the methods to different benchmarks, providing information on how these values are determined, either empirically or automatically, is important.

**Questions:**

1. From how humans manage memory, it seems there should be some transition between/from fine-grained memory to coarse-grained memory, yet that is not presented in the method. Why? Both intuitively and empirically.

2. How to ensure the quality of the synthesized training data for AgentFold. While the method is proposed to emulate the human memory consolidation process, it is unclear if the synthesized data ensures this critical property.

---

> ### Author Response · Authors · 2025-11-21
>
> Thanks for your time. We are glad to see that you ackowledge the importance of the problem, the novelty of our approach, and the experimental improvements. We would like to address your remaining concerns in the following.
>
> &nbsp;
>
> ---
>
> **W1:** Lack of Clarity in Method Description. While Section 3 describes the two condensation methods at a high level, it is still unclear what counts as a “fine-grained” summary (in granular condensation) and what counts as a “coarse-grained” summary (dense condensation). It would be helpful if Figure 2 (or an additional figure) could show a few concrete examples of both.
>
> **Response:**
>
> Thanks for the advice. We apologize for the lack of concrete examples in the main text. We will move the illustrative cases from the Appendix (Fig. 6 & 8) to the main paper in the revision. To clarify immediately:
>
> - **Fine-grained summary (granular condensation)** refers to case where agent decides to only `fold` the latest interaction while keeping the remaining context the same (i.e., folding one single step).
> - **Coarse-grained summary (deep consolidation)** refers to case where agent decides to `fold` the latest intereaction together with several previous steps (i.e., folidng several steps).
>
> Please refer to our descriptions from Line 232 to Line 242. Also, please refer to Figure 6 and Figure 8, those ones in the pattern of [Compressed Step x] are `fine-grained` and those ones in the pattern of [Compressed Step x to y] `coarse-grained `.
>
> We will add a dedicated Figure X in Section 3 explicitly contrasting these inputs and outputs side-by-side. Thanks for the advice.
>
> &nbsp;
>
> ---
>
>
>
> **W2:** **Lack of Hyper-Parameter Choices.** Several values could be critical to the empirical performance of the proposed method, but are not reported, justified, or hard to find. These values include: (i) number of steps to trigger consolidation, (ii) number of fine- and coarse-grained memory entries to maintain, (iii) number of steps of the “latest interaction”. Further, if these values need to be adjusted when applying the methods to different benchmarks, providing information on how these values are determined, either empirically or automatically, is important.
>
> **Responses:**
>
> Sorry for the confusion. Please allow us to clarify this. First of all, we would like to emphasize a core advantage of AgentFold: **The system is completely hyper-parameter-free during inference.**
>
> &nbsp;
>
> In the widely-used paradigm ReAct, at each step, the agent's response has two key components: `thinking` and `action`, which results in a block of `observation` (defined as `latest interaction` in this paper to distinguish it with folded observations). Both the `thinking` and `action` are proactively generated and determined by the agent itself, without human intervention or hyper-paramter control.
>
> Similarly, our AgentFold is an agentic paradigm where agent generates four components `thinking`, `folidng`, `motivation`, `action`, all of which are **proactively determined by the agent itself, without human intervention or hyper-paramter control.** That is, at every step, the agent would **trigger the `folding` operation, where whether fine- or coarse-grained `folding` would be automatically determined by the agent** (i.e., either folds one or multiple steps). And at each step, the most recent `motivation `, `action` and `observation` from the environment resulted by this `action` are combined and presented in the context as `latest intereaction`.
>
> &nbsp;
>
> **Because there are no hard-coded hyperparameters, no tuning is needed when transferring to different benchmarks.** The agent dynamically adjusts its context management strategy based on the task complexity itself. We will make this "Parameter-Free" property explicit in the methodology section to prevent future confusion.
>
>
> &nbsp;
>
> ---
>
> Overall, we hope that our responses can fully address your concerns and will be grateful for any feedback.

---

### Official Review · Reviewer_nh9H · 2025-10-30

**Soundness:** 2
**Presentation:** 3
**Contribution:** 3
**Rating:** 4
**Confidence:** 3

**Summary:**

This paper proposes AgentFold, an agent for information seeking tasks which proactively manages its context, enabling more efficient processing and efficacy for long-horizon tasks. The context management is achieved via context folding approach which leverages two types of operations, 1) Granular Condensation that preserves fine-grained details of single steps, 2) Deep Consolidation that merges multiple steps into coarse summaries. To train the model to effectively leverage folding, a dataset is generated using an LLM, which the final model is tuned on with SFT. The experimental results on information-seeking benchmarks performance show matching or exceeding much larger models, while maintaining efficient context usage.

**Strengths:**

- The proposed approach demonstrates strong empirical results, consistent across several benchmarks and compared against several baseline approaches.
- The problem of context explosion in ReAct agents is well-motivated, and the design of folding mechanisms are intuitive.
- The proposed approach shows significantly enhanced context efficiency, and shows better long-horizon scaling properties.

**Weaknesses:**

- Missing details of Fold Generator: It is not specified which LLM is used, which prompts were used, and how many trajectories were generated and leveraged for the training data generation. Without these details, it's difficult to assess whether the improvements come from the proposed folding method or superior training data.
- Lack of ablations: Similarly, the authors do not provide any ablation studies of folding components, and it is difficult to understand how the proposed components contribute to performance.

**Questions:**

- Have the authors tried adding the folding mechanism to baseline and proprietary LLMs?
- What are the failure modes of the proposed approach? Are there cases where folding can impact performance due to information loss?
- Is the approach generalizable to other long-horizon tasks, or applicable to other web automation tasks such as WebArena?

---

> ### Author Response · Authors · 2025-11-21
>
> Thanks for your time. We are glad that you acknowledge our performance, method, and intuition. We would like to address your remaining concerns in the following.
>
> &nbsp;
>
> ---
>
> **W1:** (1) Missing details of Fold Generator: It is not specified which LLM is used, which prompts were used, and how many trajectories were generated and leveraged for the training data generation.
>
> (2) Without these details, it's difficult to assess whether the improvements come from the proposed folding method or superior training data.
>
> **Response:**
>
> Thanks for the advice and sorry for the missing details. We would like to provide more details here and elaborate more in the next revision. We could answer from two perspectives.
>
> **(1) Implementation Details.** Within Fold-Generator, multiple open-source LLMs are leveraged to enhance diversity. The QA set is split into two, whose trajectories are generated by GLM-4.5 and DeepSeek-V3.1 respectively. Within the tool for web visiting, Qwen3-30B-A3B is leveraged for interpreting the html content from web url. The key components in the prompt is the following (more details would be included in the revision):
>
> - Role and Goal: Defines the AI as a "Meticulous Reasoning Agent" tasked with solving a Question.
> - Critical Output Structure: A mandatory, three-part response format:
>   1. A `<folding>` block (to summarize history).
>   2. A `<motivation>` block (to explain the thought process).
>   3. A `<tool_call>` block or `<answer>` block
> - Guiding Principles: Core rules for the agent's behavior, including to fold history smartly, analyze deeply, and verify thoroughly.
>
> Approximately, 30k trajectories are generated and after filtering 20k are leveraged for training.
>
> &nbsp;
>
> (2) **Method > Data.** Improvements come from the proposed folding method or superior training data? We believe that there might be some misunderstandings that we could clarify.
>
> We strongly argue for the folding mechanism based on this key observation: **our final agent surpasses the performance of represnetative open-source agentic LLMs, including those we leveraged to generate training data**. For example, on BrowseComp, AgentFold (30B) achieves **36.2**, DeepSeek-V3.1 achieves 30.0, GLM-4.5 achieves 26.4. If one leverage the same QA set for distillation, the resulting performance would be lower than 30.0 (probably much lower). This observation could strongly verify that the benefits brought by our proposed folding mechanism, which enable the agents to manage long horizons better.
>
>
> &nbsp;
>
>
> ---
>
>
>
> **W2:** Lack of ablations: Similarly, the authors do not provide any ablation studies of folding components, and it is difficult to understand how the proposed components contribute to performance.
>
> **Response:**
>
> Thanks for pointing this out. We could answer this from two perspectives.
>
> (1) First, following your advice, we conduct an experiment to verify the performance of AgentFold without leveraging the `motivation` block.
>
> | Method | AgentFold (motivation removed) | AgentFold (motivation replaced by thinking) | AgentFold |
> | ------ | ------------------------------ | ------------------------------------------- | --------- |
> | Acc    | 26.3                           | 33.5                                        | 36.7      |
>
> From the table, we see that the motivation block is also a critical component besides the folding block. This is reasonable: if we remove the motivation block in the context, then the agent cannot clearly know the logical relationships among tool calls, making it hard to plan clearly and logically; if we replace the motivation with the thinking part, then since the thinking could be lengthy and noisy, the performance would be affected.
>
> &nbsp;
>
> (2) Second, the reason why we did not conduct an ablation study is that the key design of our paper is the introduced `folding` operation, a simple and general method that few additional designs are introduced and can be ablated. The motivation block is an component associated with (based on) folding operation since for better context management, agent should know the cause and effect of previous steps (i.e., motivation) so that it could plan better (for both folding and tool calling). So, previously, we regarded these two components as a whole, yet, we now split the motivation block apart to show its effects in the above table following your suggestion.
>
> &nbsp;
>
> ---
>
> Overall, we hope that our responses can fully address your concerns and will be grateful for any feedback.

---

### Official Review · Reviewer_ZzZg · 2025-11-01

**Soundness:** 3
**Presentation:** 3
**Contribution:** 3
**Rating:** 6
**Confidence:** 4

**Summary:**

This paper introduces AgentFold, a web agent architecture designed to handle long-horizon tasks by proactively managing its interaction history. Instead of either storing the entire trajectory (ReAct-style) or repeatedly summarizing it at every step (which risks losing crucial details), AgentFold performs learned context folding. It decides when to keep high-resolution details, when to condense recent actions into smaller summaries, and when to merge longer sub-trajectories into coarse abstractions. This yields a multi-scale memory representation that scales more gracefully with task length. Experiments are conducted on BrowseComp and BrowseComp-ZH, where the approach shows state-of-the-art performance among open-source agents of comparable and even substantially larger parameter sizes. The approach is technically sound and frames its memory representation rigorously. This paper presents an idea is both intuitive and practically impactful, and the experimental results substantiate its benefits. Some further analysis, like training pipeline cost and folding behavior interpretability, would strengthen the contribution.

**Strengths:**

1. The idea of proactively managing its interaction history and dynamic consolidation of interactions is intuitive.
2. The results on BrowseComp and BrowseComp-zh are promising.

**Weaknesses:**

1. There is no discussion of the cost of the folding procedure and its associated overhead.
2. The evaluation on BrowseComp* is rather limited. How about adding this folding mechanism to web agents in general and evaluating on Online-Mind2Web?

**Questions:**

Have the authors considered adding this folding mechanism to web agents in general and evaluating it on Online-Mind2Web?

---

> ### Author Response · Authors · 2025-11-21
>
> We thank the reviewer for the time and the positive assessment, particularly for recognizing our proposed memory mechanism as 'intuitive,' 'technically sound,' and 'practically impactful.' We appreciate the constructive suggestions regarding cost analysis and evaluation scope, which we address below.
>
> &nbsp;
>
> ---
>
> **W1:** There is no discussion of the cost of the folding procedure and its associated overhead.
>
> **Response:**
>
> Thanks for the advice. We would like to address this from two perspectives: token generation cost and computational trade-off.
>
> **(1) Token Generation Overhead (Low):** Following your advice, we measure the token consumption of the folding operation in the full output tokens.
>
> &nbsp;
>
> | Part  | Thinking | Folding+Motivation | Tool Call |
> | :------: | :------: | :------: | :------: |
> | Ratio in full output tokens | 92.6%    | **5.39%**              | 2.01%     |
>
> &nbsp;
>
> From the table, we can see that our folding operation together with the associated motivation part **only consume 5.39%**. This indicates that the generation overhead is marginal compared to the core reasoning process.
>
> **(2) Computational Trade-off (Positive):** While folding incurs a small generation cost (~5% extra output tokens), it significantly **reduces the input context length (more compact and less noisy)** for subsequent steps (from linear growth to sub-linear). In long-horizon tasks (e.g., 100+ steps), the savings in input processing cost (pre-filling) and attention computation far outweigh the marginal cost of generating folding tokens. This makes AgentFold efficient and effective than ReAct baselines in long trajectories.
>
> &nbsp;
>
> ---
>
> **W2:** The evaluation on BrowseComp* is rather limited. How about adding this folding mechanism to web agents in general and evaluating on Online-Mind2Web?
>
> **Response:**
>
> Thanks for the advice. We would like to discuss with you from two perspetives.
>
> (1) **Conceptually and methodmadically, our AgentFold is a general mechanism (like ReAct) that is applicable to general scenarios including Online-Mind2Web.** Compared to ReAct (also a general mechanism), our core innovation lies in our introduced folding operation in addition to standard operation such as thinking and action. This operation is designed to manage the context of LLMs, which is orthogonal to tasks, scenarios, and modality. Therefore, our AgentFold is a general method as ReAct.
>
> (2) Following your advice, we were planning to evaluate on Online-Mind2Web. However, we noticed that to correctly evaluate on this benchmark, we need to introduce a new modality (image) and a new set of tools, collect training data from scratch. This would be **too resource- and time-consuming**, making it hard to make it during the rebuttal period. We really need your understanding on this part, as LLM and MLLM themselves are generally two distinct directions in the research community.
>
> (3) **Evidence of Generalization (WideSearch & GAIA): To address the concern about "limited evaluation," we respectfully point out that beyond BrowseComp and BrowseComp-ZH, we have also evaluated AgentFold on WideSearch and GAIA**. BrowseComp and BrowseComp-ZH are two benchmarks that focus on testing the capabilities of agents in finding hard-to-find information, which cover two common languages in the world. Unlike these, WideSearch is a benchmark that evaluates agent reliability on large-scale collection tasks (e.g., collecting data and forming a 100-row table), GAIA is a benchmark for evaluating agent's general capabilities, covering diverse scenarios beyond searching.
>
> Overall, we believe that AgentFold is a general mechanism that can be applied to many other scenarios. We also acknowledge that extending it to more broad directions such as introducing more modalities and more tools is a worth-exploring future direction.
>
> &nbsp;
>
> ---
>
> Overall, we hope that our responses can fully address your concerns and will be grateful for any feedback.

---

### Official Review · Reviewer_C2c8 · 2025-11-02

**Soundness:** 3
**Presentation:** 3
**Contribution:** 3
**Rating:** 8
**Confidence:** 3

**Summary:**

This paper introduces AgentFold, a new paradigm for long-horizon web agents that addresses a persistent trade-off in context management: ReAct-style agents accumulate noisy, ever-growing logs, while fully summarizing agents lose key details prematurely. AgentFold proposes a proactive “context folding” mechanism inspired by human cognitive consolidation—learning when to condense or merge past interactions into structured multi-scale summaries. The system dynamically curates its own context via two operations—granular condensation (folding a single step) and deep consolidation (folding multiple steps). It is trained through a specialized Fold-Generator pipeline producing structured reasoning trajectories, and fine-tuned on Qwen3-30B. Across benchmarks like BrowseComp, BrowseComp-ZH, and WideSearch, AgentFold achieves strong performance (36.2%, 47.3%, and 62.1% respectively), outperforming much larger open-source and even proprietary agents such as OpenAI’s o4-mini, while maintaining sublinear context growth over hundreds of steps

**Strengths:**

The conceptual clarity and human-analogy framing are exceptional. The authors identify context saturation as a bottleneck and present a clean, modular solution that integrates memory curation into the agent’s reasoning loop. The paper demonstrates strong experimental rigor: detailed comparisons with major baselines (DeepSeek-V3.1-671B, GLM-4.5-355B, WebSailor, etc.), concrete scaling analyses, and insightful case studies that show the agent’s ability to self-reflect and prune unproductive trajectories. The writing is polished, the methodology well-motivated, and the results both impressive and interpretable. The sub-linear token and block growth analyses convincingly validate the practical efficiency of the folding mechanism.

**Weaknesses:**

Overall solid paper. No significant weakness.

**Questions:**

N/A

---

> ### Author Response · Authors · 2025-11-21
>
> Thanks for your time and appreciation. We are delighted that the key concepts we aimed to convey—specifically the proactive context folding mechanism and its efficiency—resonated well with you.
>
> To further strengthen the work, we have absorbed suggestions from the broader review process. In the revision, we plan to present more technical details, add supplementary ablation studies, and provide further analytical results.

---

### Meta-Review · Area_Chair_izDA · 2026-01-06

**Summary:**

The paper presents a method for an LLM-based agent to condition on, and add to, a summary of its interaction history, rather than needing to condition directly on all past reasoning, actions, and observations in the history (as in a standard ReAct agent). A Qwen-3-30B-A3B model is fine-tuned to have this behavior, using distillation from larger LLMs (the author response indicates these are GLM-4.5 and DeepSeek-V3.1) with rejection sampling to generate the training data using questions from the recent WebSailor work. The resulting model is then validated on the BrowseComp, BrowseComp-ZH, WideSearch, and GAIA benchmarks, where it obtains higher performance than most other works on these benchmarks, and shows substantially better token efficiency than the ReAct baseline.

Strengths: The proposed approach is well-motivated, and the reviewers found it to be an intuitive and clean solution to the problem of context saturation. The context token savings compared to the ReAct baseline is pretty substantial (but see below for a concern about caching). Despite this savings, the performance is still very strong, outperforming prior work on these benchmarks.

**Reviewer Concerns:**

The major concerns of the reviewers were mostly addressed in the author response, but there are still a few remaining (including ones that I the AC had that reviewers didn't raise):

Unaddressed concerns:
- Nearly all details of the training data generation ("Fold Generator") are missing in the paper. Some of these are given, helpfully, in the response to nh9H, but more should be said about the rejection sampling criteria, the prompts for the models, and the tasks in the QA training set.
- The paper uses Qwen-3.1-30B-A3B-Instruct as a base model, so it should report results from it in Table 1.
- I was unclear on why the fine-tuned model can outperform either the GLM-4.5 and DeepSeek-V3.1 models that it distills from. It would help to do the experiments outlined in the response to nh9H (leveraging the same QA set for distillation) to see how much of the benefit is from distillation and how much from the context folding.
- I wasn't totally clear on the ablation experiment in nh9H response W2.
- Although no reviewer raised this, the ReAct baseline can benefit substantially from KV caches to reuse quite a bit of the computation in each turn, since the history is strictly being appended to. In the proposed approach, elements of the history can be replaced, which could invalidate the use of the KV cache. It would make the paper much stronger to show wall clock times to compare AgentFold vs ReAct in an optimized framework which uses KV caching, such as VLLM.

**Reviewer Scores:**

I expect mj6d might have raised their score to a 6, so that the final scores might be 8 / 6 / 6 / 4.

---

### Decision · Program_Chairs · 2026-01-26

Accept (Poster)